# The mAB 13A4 monoclonal antibody to the mouse PROM1 protein recognizes a structural epitope

**Fatimah Matalkah**[1], **Scott Rhodes**[1], **Visvanathan Ramamurthy**[1,2], **Peter Stoilov**[1]*

**1** Department of Biochemistry and Molecular Medicine, Robert C. Byrd Health Sciences Center, West Virginia University, Morgantown, West Virginia, United States of America, **2** Department of Ophthalmology and Visual Sciences, Robert C. Byrd Health Sciences Center, West Virginia University, Morgantown, West Virginia, United States of America

* pstoilov@mix.wvu.edu

**Data Availability Statement:** All relevant data are within the manuscript and its Supporting Information files.

**Funding:** This study was supported by the National Institutes of Health, National Eye Institute (https://

## Abstract

PROM1 (CD133, AC133) is a protein that is required for the maintenance of primary cilia. Mutation in the *Prom1* gene in humans and animal models are associated with several forms of retinal degeneration. mAB 13A4 is the main reagent used to detect the mouse PROM1 protein. We endeavored to map the epitope of the rat monoclonal antibody mAB 13A4 to the mouse PROM1 protein. Deletion mutagenesis demonstrated that mAB 13A4 recognizes a structural epitope that is stabilized by two of the extracellular domains of PROM1. Furthermore, the affinity of mAB 13A4 to the major PROM1 isoform in photoreceptor cells is significantly reduced due to the inclusion of a photoreceptor-specific alternative exon in the third extracellular domain. Interestingly, a deletion in the photoreceptor specific isoform of six amino acids adjacent to the alternative exon restored the affinity of mAB 13A4 to PROM1. The results of the mutagenesis are consistent with the computationally predicted helical bundle structure of PROM1 and point to the utility of mAB 13A4 for evaluating the effect of mutations on the PROM1 structure. Our results show that the PROM1 isoform composition needs to be considered when interpreting tissue and developmental expression data produced by mAB 13A4.

## Introduction

Prominin-1 (PROM1, CD133, AC133) was identified as the antigen of monoclonal antibodies raised against human hematopoietic progenitors and mouse neuroepithelial cells [1–3]. PROM1 is a glycosylated membrane protein with five transmembrane and three extracellular domains [1, 2]. PROM1 is a member of a conserved family of proteins involved in modulating the architecture of cellular protrusions, such as microvilli and cilia [4–6]. Mutations in the human PROM1 gene have been reported in cases of Retinitis Pigmentosa, Stargardt Disease, Macular Dystrophy, Leber Congenital Amaurosis and are primarily associated with cone-rod dystrophy [7–11]. The forms of retinal degeneration associated with PROM1 mutations represent a diverse spectrum in terms of disease onset and severity of the presented symptoms.

www.nei.nih.gov/) in the form of a grant to PS and VR [2R01EY025536], the West Virginia University Health Sciences Center Office of Research and Graduate Education (https://health.wvu.edu/research-and-graduate-education/) in the form of a grant to PS, and the National Institute of General Medical Sciences (NIGMS) Visual Sciences Center of Biomedical Research Excellence (VS-CoBRE) (https://www.nigms.nih.gov/) in the form of a grant to VR [P20GM144230]. The funders had no role in study design, data collection and analysis, decision to publish, or preparation of the manuscript.

**Competing interests:** The authors have declared that no competing interests exist.

Animal models lacking *Prom1* or expressing the dominant Arg373Cys mutant recapitulate the retinal degeneration phenotype and display defects in disk morphogenesis [9, 12, 13]. Most mutations in *Prom1* are recessive and result in loss of function due to premature stop codons. Notably, three missense mutations, Leu245Pro, Arg373Cys, and Asp829Asn, have dominant inheritance patterns [7]. The mechanisms underlying the pathology of PROM1 mutations and the reasons for the confinement of their phenotypes to the visual system are still unclear. Recent evidence of genetic interaction with ABCA4 points to a role in transport of all-trans retinaldehyde across the photoreceptor disk membranes [14].

The *Prom1* gene produces multiple splicing isoforms that can be tissue, and cell type specific [15–19]. Six alternative exons in the mouse *Prom1* can potentially produce 24 splice variants, although to date only eight have been enumerated [15, 16]. In mouse photoreceptor cells a microexon introduces 6 amino acids in the photoreceptor-specific s8 isoform of *Prom1* (Fig 1A) [16, 17]. This exon, referred to as exon 19 to denote its position in the coding region of the reference clone BC028286, is located at chr5:44,012,499–44,012,517 in the GRCm39 assembly of the mouse genome [15]. While exon 19 is present in most vertebrate clades, it is not used in the primates, including humans, due to mutations that disrupt either the 3' or the 5' splice site of the exon (Fig 1B).

Despite playing conserved and critical functions the proteins encoded by the Prominin-1 genes show relatively low conservation of their primary amino acid sequence. For example, human and mouse PROM1 proteins share approximately 61% sequence identity. Consequently, antibodies to PROM1 tend to be species specific. In mice, the rat monoclonal antibody mAB 13A4 is the reagent typically used to detect PROM1. The mAB 13A4 antibody was raised against extract from mouse neuroepithelium [2]. Its antigen was cloned by screening for reactivity with mAb 13A4 of a phage library of mouse kidney cDNA [2]. mAB 13A4 is speculated to recognize part of the third extracellular domain of PROM1 because of a truncating mutation in that domain that abolishes its binding, but the epitope was never mapped [8, 20].

Prompted by a discrepancy between the PROM1 protein levels measured in the postnatal retina by mAB 13A4 and the mouse monoclonal antibody to PROM1 ab27699, we set out to map the epitope of mAB 13A4. We found that mAB 13A4 recognizes a structural epitope that can be disrupted by deletions in the second and third extracellular domains. Furthermore, the affinity of mAB 13A4 to PROM1 is dramatically reduced by the inclusion of the photoreceptor-specific exon 19.

## Materials and methods

### Animals

The *Prom1*^rd19^ mice were acquired from the Jackson laboratory (B6. BXD83-Prom1^rd19^/Boc, Stock No: 026803). All experiments were conducted with the approval of the Institutional Animal Care and Use Committee at West Virginia University.

### *Prom1* clones

We obtained a full length Mammalian Gene Collection clone of the mouse photoreceptor specific isoform (s8) of *Prom1* from Horizon Discovery (clone ID: 4502359, NCBI accession: BC028286). Gibson assembly (NEB# E5510) was used to generate Flag-tagged *Prom1* clones and deletion mutants in pcDNA3.1+ (Invitrogen). All clones are deposited at AddGene (https://www.addgene.org/Peter_Stoilov/). The primers used for cloning are listed in S1 Table.

**Cell culture and transfection.** Mouse Neuro-2a (N2a) neuroblastoma cells (ATCC CCL-131) were cultured in OptiMEM reduced serum media buffered with sodium bicarbonate and supplemented with 4% (v/v) fetal bovine serum (FBS, R&D Systems, Minneapolis, MN). The

## A

```
s8  1    MALVFSALLLLGLCGKISSEGQPAFHNTPGAMNYELPTTKYETQDTFNAGIVGPLYKMVHIFLNVVQPND
s2  1    MALVFSALLLLGLCGKISSEGQPAFHNTPGAMNYELPTTKYETQDTFNAGIVGPLYKMVHIFLNVVQPND

s8  71   FPLDLIKKLIQNKNFDISVDSKEPEIIVLALKIALYEIGVLICAILGLLFIILMPLVGCFFCMCRCCNKC
s2  71   FPLDLIKKLIQNKNFDISVDSKEPEIIVLALKIALYEIGVLICAILGLLFIILMPLVGCFFCMCRCCNKC

s8  141  GGEMHQRQKQNAPCRRKCLGLSLLVICLLMSLGIIYGFVANQQTRTRIKGTQKLAKSNFRDFQTLLTETP
s2  141  GGEMHQRQKQNAPCRRKCLGLSLLVICLLMSLGIIYGFVANQQTRTRIKGTQKLAKSNFRDFQTLLTETP

s8  211  KQIDYVVEQYTNTKNKAFSDLDGIGSVLGGRIKDQLKPKVTPVLEEIKAMATAIKQTKDALQNMSSSLKS
s2  211  KQIDYVVEQYTNTKNKAFSDLDGIGSVLGGRIKDQLKPKVTPVLEEIKAMATAIKQTKDALQNMSSSLKS

s8  281  LQDAATQLNTNLSSVRNSIENSLSSSDCTSDPASKICDSIRPSLSSLGSSLNSSQLPSVDRELNTVTEVD
s2  281  LQDAATQLNTNLSSVRNSIENSLSSSDCTSDPASKICDSIRPSLSSLGSSLNSSQLPSVDRELNTVTEVD

s8  351  KTDLESLVKRGYTTIDEIPNTIQNQTVDVIKDVKNTLDSISSNIKDMSQSIPIEDMLLQVSHYLNNSNRY
s2  351  KTDLESLVKRGYTTIDEIPNTIQNQTVDVIKDVKNTLDSISSNIKDMSQSIPIEDMLLQVSHYLNNSNRY

s8  421  LNQELPKLEEYDSYWWLGGLIVCFLLTLIVTFFFLGLLCGVFGYDKHATPTRRGCVSNTGGIFLMAGVGF
s2  421  LNQELPKLEEYDSYWWLGGLIVCFLLTLIVTFFFLGLLCGVFGYDKHATPTRRGCVSNTGGIFLMAGVGF

s8  491  GFLFCWILMILVVLTFVVGANVEKLLCEPYENKKLLQVLDTPYLLKEQWQFYLSGMLFNNPDINMTFEQV
s2  491  GFLFCWILMILVVLTFVVGANVEKLLCEPYENKKLLQVLDTPYLLKEQWQFYLSGMLFNNPDINMTFEQV

s8  561  YRDCKRGRGIYAAFQLENVVNVSDHFNIDQISENINTELENLNVNIDSIELLDNTGRKSLEDFAHSGIDT
s2  561  YRDCKRGRGIYAAFQLENVVNVSDHFNIDQISENINTELENLNVNIDSIELLDNTGRKSLEDFAHSGIDT

s8  631  IDYSTYLKETEKSPTEVNLLTFASTLEAKANQLPEGKLKQAFLLDVQNIRAIHQHLLPPVQQSL**KFVRVR**
s2  631  IDYSTYLKETEKSPTEVNLLTFASTLEAKANQLPEGKLKQAFLLDVQNIRAIHQHLLPPVQQSL------

s8  701  NTLRQSVWTLQQTSNKLPEKVKKILASLDSVQHFLTNNVSLIVIGETKKFGKTILGYFEHYLHWVFYAIT
s2  695  NTLRQSVWTLQQTSNKLPEKVKKILASLDSVQHFLTNNVSLIVIGETKKFGKTILGYFEHYLHWVFYAIT

s8  771  EKMTSCKPMATAMDSAVNGILCGYVADPLNLFWFGIGKATVLLLPAVIIAIKLAKYYRRMDSEDVYDD--
s2  765  EKMTSCKPMATAMDSAVNGILCGYVADPLNLFWFGIGKATVLLLPAVIIAIKLAKYYRRMDSEDVYDDVE

s8  839  ----------------------------PSRY 842
s2  835  TVPMKNLEIGSNGYHKDHLYGVHNPVMTSPSRY 867
```

## B

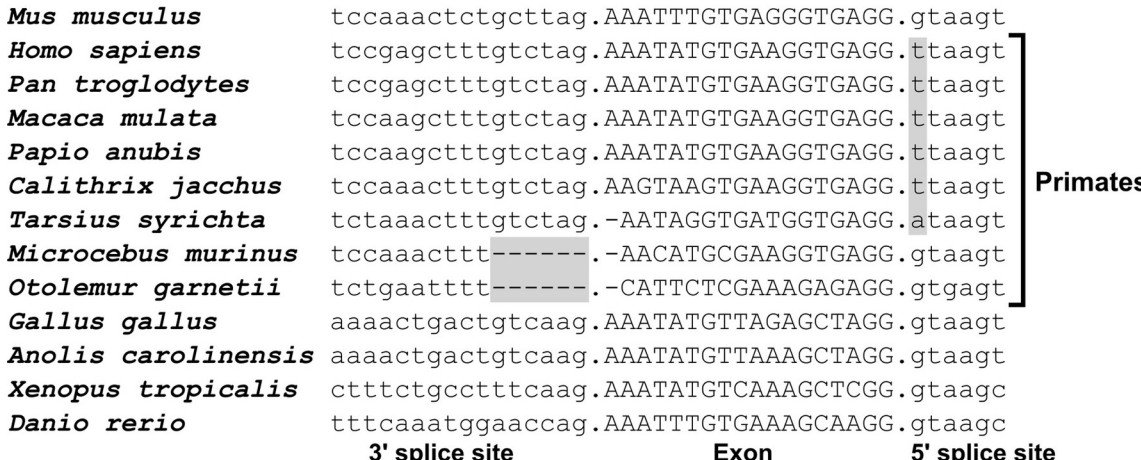

```
Mus musculus       tccaaactctgcttag.AAATTTGTGAGGGTGAGG.gtaagt
Homo sapiens       tccgagctttgtctag.AAATATGTGAAGGTGAGG.ttaagt  ┐
Pan troglodytes    tccgagctttgtctag.AAATATGTGAAGGTGAGG.ttaagt  │
Macaca mulata      tccaagctttgtctag.AAATATGTGAAGGTGAGG.ttaagt  │
Papio anubis       tccaagctttgtctag.AAATATGTGAAGGTGAGG.ttaagt  │
Calithrix jacchus  tccaaactttgtctag.AAGTAAGTGAAGGTGAGG.ttaagt  │ Primates
Tarsius syrichta   tctaaactttgtctag.-AATAGGTGATGGTGAGG.ataagt  │
Microcebus murinus tccaaacttt------.-AACATGCGAAGGTGAGG.gtaagt  │
Otolemur garnetii  tctgaatttt------.-CATTCTCGAAAGAGAGG.gtgagt  ┘
Gallus gallus      aaaactgactgtcaag.AAATATGTTAGAGCTAGG.gtaagt
Anolis carolinensis aaaactgactgtcaag.AAATATGTTAAAGCTAGG.gtaagt
Xenopus tropicalis ctttctgcctttcaag.AAATATGTCAAAGCTCGG.gtaagc
Danio rerio        tttcaaatggaaccag.AAATTTGTGAAAGCAAGG.gtaagc
                      3' splice site         Exon        5' splice site
```

**Fig 1. Photoreceptor specific splice variant of PROM1. A)** Alignment of the photoreceptor-specific s8 isoform of PROM1 (RefSeq NP_001157057) to the ubiquitously expressed isoform s2 (RefSeq NP_001157049). The amino acids encoded by the photoreceptor specific exon 19 are shown in bold and shaded in gray. The third extracellular domain is underlined. **B)** Alignment of vertebrate exon 19 sequences including the adjacent 3' and 5' splice sites. Mutations inactivating the splice sites in primates are shaded in gray.

cells were grown at 37˚C in a 5% CO2 humidified atmosphere. The cDNA clones were transiently transfected in N2a cells using polyethylenimine [21]. Cell lysates for Western blot analysis were collected at 24 hours post-transfection.

## Denaturing gel electrophoresis and western blot

Flash-frozen mouse retinas and N2a cells transiently transfected with Prom1-expressing constructs were lysed in RIPA buffer (50 mM Tris HCl-PH 8.0, 150 mM NaCl, 1.0% TritonX-100, 0.5% Sodium Deoxycholate, 0.1% SDS) supplied with protease (Sigma-Aldrich catalog# 535140-1ML) and phosphatase inhibitors cocktail (Sigma-Aldrich catalog # P5726-1 ML). After homogenization, the lysate was incubated on ice for 10 mins, then cleared by centrifugation for 15 mins. Where applicable glycosylation was removed by treating with deglycosylation MixII (NEB# P6044) following the manufacturer's recommendation. 20 μg of protein extract was resolved in 4–20% polyacrylamide SDS–PAGE gel and transferred onto polyvinylidene difluoride (PVDF) membranes (Immunobilon-FL, Millipore). After blocking with BSA in PBST (Phosphate- buffered saline with 0.1% Tween-20), the membranes were probed with primary antibodies overnight at 4˚C, followed by incubation with fluorescently labeled (Alexa Fluor 647 or 488, Jackson ImmnuoResearch) secondary antibodies for 1 hour at room temperature. The membranes were then imaged on Amersham Typhoon Phosphorimager (GE Healthcare).

Serial dilution was performed to ascertain the linearity of western blot quantification (S1 Fig). Lysates of cells transfected with the s8(-ex19) clone were diluted with extracts from cells transfected with an empty vector to maintain equal loading. The lysates were then probed on western blot by mAB 13A4 and imaged as described above. Linear regression was performed in Graphpad Prism on the scaled and normalized band intensities.

## Blue Native Polyacrylamide Gel Electrophoresis (BN PAGE)

The samples were lysed in BN PAGE sample buffer (Thermo Fisher Catalog# BN2008) containing 1% digitonin and protease inhibitors following the manufacturer's recommendation. The lysates were treated with benzonase at room temperature for 30 minutes to shear the DNA, and cleared by centrifugation for 15 mins. Prior to electrophoresis, the samples were mixed with Coomassie G-250 and resolved in 3–12% NativePAGE Bis-Tris gel (Invitrogen Catalog #BN1003BOX) as per the manufacturer's recommendations. The gels were then transferred on PVDF membranes (Immunobilon-FL, Millipore) following the manufacturer's recommendations. After transfer, the membranes were incubated in 8% acetic acid for 15 minutes to fix the proteins, rinsed with deionized water, and air-dried. The lane containing the size standard was cut from the membrane and stained with Ponceau S, and imaged on Amersham Typhoon Phosphorimager (GE Healthcare) using the Cy2 filter set. The membranes were blocked, probed with antibodies, and imaged as described above in the denaturing gel electrophoresis protocol.

**Western blot quantification.** Western blot images were analyzed using ImageQuant TL (version 8.1, GE Healthcare). Band intensities in denaturing gel blots were measured after subtracting the background using the rolling ball algorithm built in the ImageQuantTL software. To enable comparison across blots, the band intensities were scaled to the total signal intensity for each blot. The signal was then normalized to the scaled intensities of the loading control.

In native gels, applying the rolling ball background subtraction algorithm and measuring individual band intensities proved unreliable due to smearing. Instead we used the signal from an adjacent lane containing extract from cells transfected with an empty vector to determine

the background and quantified the total signal intensity above 240 Kda within each well. The data was then processed as described above for the denaturing gels.

**Antibodies.** The primary and secondary antibodies that were used throughout our studies include the following: mouse anti-β-tubulin (1:5000; catalog # T8328-200ul, Sigma-Aldrich, St. Louis, MO), mouse anti-GAPDH (1:20,000; custom made), mouse anti-flag M2 (1:1000, catalog # F1804-200UG, Sigma-Aldrich, St. Louis, MO), rat anti- Prom1(1:1000, clone ID:13A4, ThermoFisher, Waltham, MA), mouse anti-GFP HRP conjugated GFP tag (1:1000, Cat #HRP-66002, ThermoFisher, Waltham, MA), mouse anti-Prom1 (1:1000, Cat #ab27699, Abcam, Cambridge, MA), Alexa Fluor 647 conjugated AffiniPure Goat-Anti rabbit IgG (1:3000, Jackson ImmunoReserach, West Grove, PA), Alexa Fluor 488 conjugated AffiniPure Goat-Anti mouse IgG (1:3000, Jackson ImmunoReserach, West Grove, PA), and Alexa Fluor 647 conjugated AffiniPure Goat-Anti rat IgG (1:3000, Jackson ImmunoReserach, West Grove, PA).

## Statistical analysis

The two-way analysis of variance (ANOVA) with Tukey Honest Significant Differences (HSD) post-hoc test or two-tailed unpaired Student's T-test were used to determine statistical significance as indicated in the results section. Quantitative data is presented as the mean of three biological replicates ±standard error of the mean. Biological replicates refer to cell lines samples grown and transfected independently, and to tissue samples collected from different animals.

**Protein structure prediction and visualization.** The RobeTTa structure prediction service was used to create models of the photoreceptor-specific s8 (RefSeq NP_001157057) isoform that contains exon 19 and the ubiquitously expressed isoform s2 (RefSeq NP_001157049) that lacks exon 19 (see Fig 1A for alignment of the two sequences). To create images of the structures we used Pymol (The PyMOL Molecular Graphics System, Version 2.0 Schrödinger, LLC).

## Results

### Discrepancy in the PROM1 protein levels measured by the mAB 13A4 and ab27699 antibodies

While investigating the expression of PROM1 in postnatal mouse retina we noticed that when measured by mAB 13A4 the PROM1 protein levels peaked at postnatal day 8, five days before the peak recorded by ab27699 (Figs 2A and 2B and S2). Furthermore, mAB 13A4 showed approximately three to five fold lower levels of PROM1 at postnatal days 13 and beyond compared to ab27699 (Fig 2A and 2B). Two way ANOVA showed significant effect of the day after birth ($F(9) = 55.26$, p-value$<2^*10^{-16}$) and the antibody used ($F(1) = 365.48$, p-value$<2^*10^{-16}$) on the measured PROM1 protein levels. According to the Tukey HSD post-hoc test the signals detected by mAB 13A4 and ab27699 were significantly different starting from postnatal day 13. It is possible that cross-reactivity of ab27699 to other proteins of size similar to PROM1 in the retina could have compromised its performance. To rule out cross-reactivity we probed retinal extracts from wild type, heterozygous, and homozygous *Prom1*$^{rd19}$ mutant retina with mAB13A4 and ab27699. The *Prom1*$^{rd19}$ allele contains a premature stop codon that abolishes the expression of the PROM1 protein [22]. Both antibodies recognized a protein just over 100KDa in size in the wild type retina that was not detected in the extract from *Prom1* knock-out retina (Fig 2C). Thus, the discrepancy in the signal between mAB13A4 and ab27699 is likely due to differences in the availability of the epitopes recognized by the two antibodies.

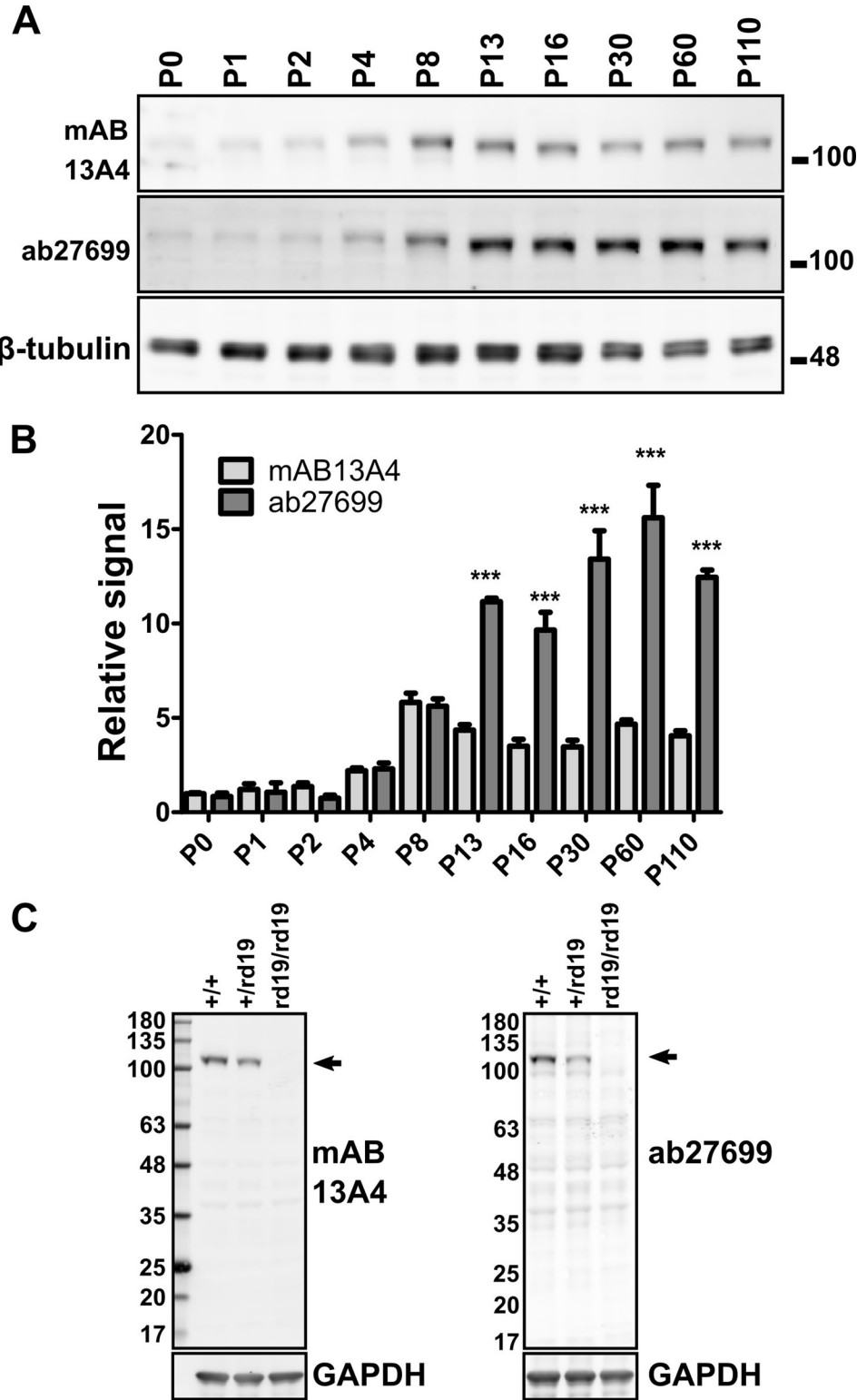

**Fig 2. Discrepancy in the levels of PROM1 as determined by the mAB 13A4 ab27699 antibodies. A)**
Immunoblotting for PROM1 in mouse retina lysate collected between postnatal day 0 (P0) and postnatal day 110
(P110) using the mAB 13A4 and the ab27699 antibodies. Anti-β-tubulin serves as a loading control. **B)** Quantification
of Prom1 level in retina lysates using the mAB 13A4 and the ab27699 antibodies. Error bars represent the standard
error of the mean (n = 3). Two-way ANOVA was used to assess the effect of the postnatal day and the antibody used

on the PROM1 signal. The statistical significance of the signal mAB 13A4 compared to ab27699 for each day was calculated by Tukey HSD post-hoc test. Tukey HSD p-values of less than 0.001 from the post-hoc test are indicated by "***". **C)** Test of the specificity of mAB 13A4 and ab27699 antibodies for detecting PROM1 using the retinal lysate from wild type mice and *Prom1^rd19* mutants that do not express PROM1 protein. Arrow indicates the position of the expected PROM1 band.

## Reduced affinity of mAB13A4 to PROM1 carrying the photoreceptor-specific exon 19

A short 18nt microexon, exon 19, is included in the *Prom1* transcripts specifically in photoreceptor cells (Fig 1) [17]. The inclusion rate of exon 19 starts to increase at postnatal day 3 and the exon 19 containing transcripts become dominant in the retina after postnatal day 8 [17]. The peptide encoded by exon 19 is inserted in the third extracellular loop of PROM1 (Fig 3A), which is also the proposed location of the mAB 13A4 epitope [8, 20]. Thus, it was possible that inclusion of exon 19 disrupts the epitope of mAB 13A4 while leaving intact the epitope of ab27699. To determine if exon 19 disrupts the mAB 13A4 epitope we generated Flag-tagged cDNA clones that either contain (s8) or skip (s8(-Ex19)) the photoreceptor specific exon 19. The cDNA were transfected in N2a cells and their expression was probed with mAB 13A4, ab27699 and anti-Flag antibodies. As expected, mAB 13A4 showed significant reduction in its affinity to the protein containing exon 19 compared to the epitope tag (Figs 3B and 3C and S3). The inclusion of exon 19 had no effect on the affinity of the ab27699. To rule out that mAB 13A4 recognizes a glycosylated epitope we treated lysates from N2a cells transfected with s8 and s8(-Ex19) expression constructs with PGNase F before probing them by western blot. Consistent with Weigmann et al. the deglycosylated proteins were recognized by the mAB 13A4 antibody (Fig 3D) [2]. The overall signal produced by mAB 13A4 was reduced when deglycosylated PROM1 was probed resulting in inconsistent detection of the s8 isoform (Figs 3D and S4).

## mAB 13A4 recognizes a structural epitope

To map the epitope of mAB13A4 we created a series of tiled deletion mutants of the s8(-Ex19) cDNA, that originated at the point where exon 19 would have been inserted and progressed in C-terminal and N-terminal direction (Fig 4A). The deletion mutants covered 108 amino acids of sequence. Nine out of ten deletions resulted in complete loss of the mAB 13A4 epitope (Figs 4B and S4). Only the most C-terminal deletion in the series (D+4) could be detected by mAB 13A4. As with the full length protein the D+4 mutant was recognized by mAB 13A4 after being deglycosylated (S4 Fig). The results of the deletion mutagenesis indicate an epitope for mAB 13A4 that is at least 94 amino acids in length. This length far exceeds the size range of linear peptide epitopes and strongly argues for a structural rather than a linear epitope [24, 25].

The western blots shown on Figs 1, 3 and 4 were performed using denaturing gel electrophoresis. Detecting a structural antigen using this approach would require the protein to renature on the membrane prior to probing with the primary antibody. Consequently, our results may reflect the propensity of the splicing isoforms and deletion mutants to renature rather than the genuine affinity of the mAB 13A4 to the native proteins. To test if this is the case we resolved the proteins produced by the s8 and s8(-Ex19) clones on native gel electrophoresis and probed the membranes with mAB 13A4 and anti-Flag antibodies. The exon 19 containing protein was recognized with reduced affinity (Figs 4C and 4D and S5) demonstrating that inclusion of exon 19 alters the structure of PROM1. The reduction in the mAB13A4 reactivity towards the s8 isoform on native gel was 3 fold compared to 8 fold reduction when denaturing gel was used to resolve the proteins. This difference likely reflects inefficient refolding of the

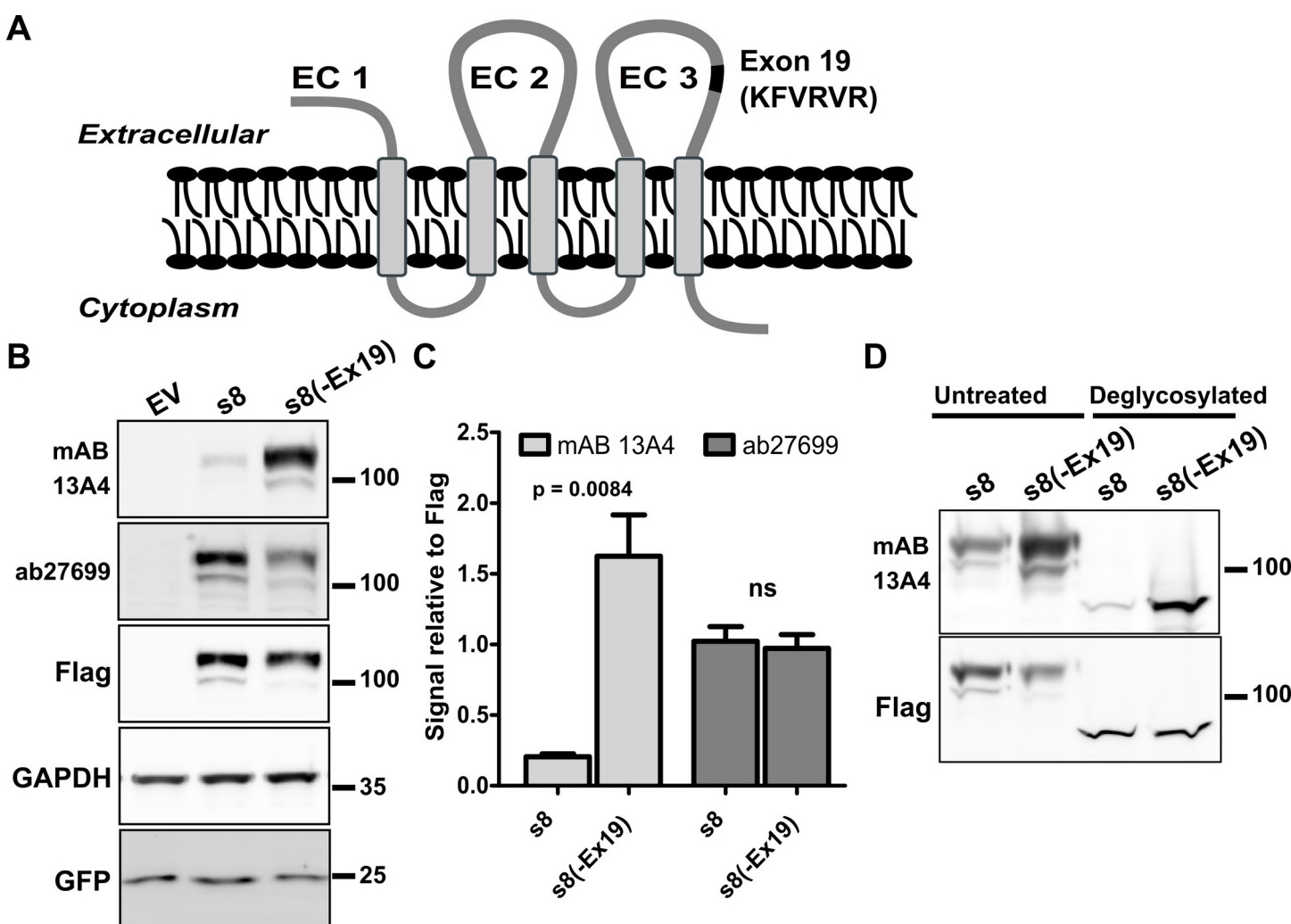

**Fig 3. The affinity of mAB 13A4 to PROM1 is affected by alternative splicing. A)** Schematic representation of the PROM1 structure showing the position of exon 19 (black) in the third extracellular domain (EC 3) of the photoreceptor specific PROM1 isoform s8 (Adapted from Corbeil et al [23]). Extracellular domains one through three are labeled as EC 1, EC 2, and EC 3 respectively. **B)** Western blot of recombinant s8 and s8 lacking exon 19, s8(-Ex19), expressed in N2a cells with mAB 13A4, ab27699 and antibody to the Flag epitope. Transfection with empty pcDNA3.1 vector (EV) was used as a negative control. All transfections were spiked with vector expressing GFP to control for transfection efficiency. **C)** Quantification of mAB 13A4 and ab27699 signals relative to the signal from the Flag-tag antibody. Error bars represent the standard error of the mean (n = 3). Unpaired Student's t-test was used to assess the statistical significance and the p-value is indicated on the chart. **D)** Deglycosylated PROM1 isoform s8 is recognized by mAB 13A4 with reduced affinity compared to protein lacking the amino acids encoded by exon 19.

PROM1 protein containing exon 19, Interestingly, under native conditions PROM1 formed higher order complexes.

## Computationally derived PROM1 tertiary structure predicts the effect of sequence manipulation on the mAB13A4 epitope

To better understand the nature of the mAB 13A4 epitope and how our deletion mutagenesis affected it we needed a tertiary structure for PROM1. There are no empirically derived structures of PROM1. Nevertheless, recent advances in computational approaches for structure prediction are producing remarkably accurate structures [26, 27]. We used the RobeTTa structure prediction service to model the structures of the mouse PROM1 isoforms s8 and s2 (Fig 1A) [26]. The photoreceptor-specific s8 isoform differs from canonical s2 isoforms by the inclusion

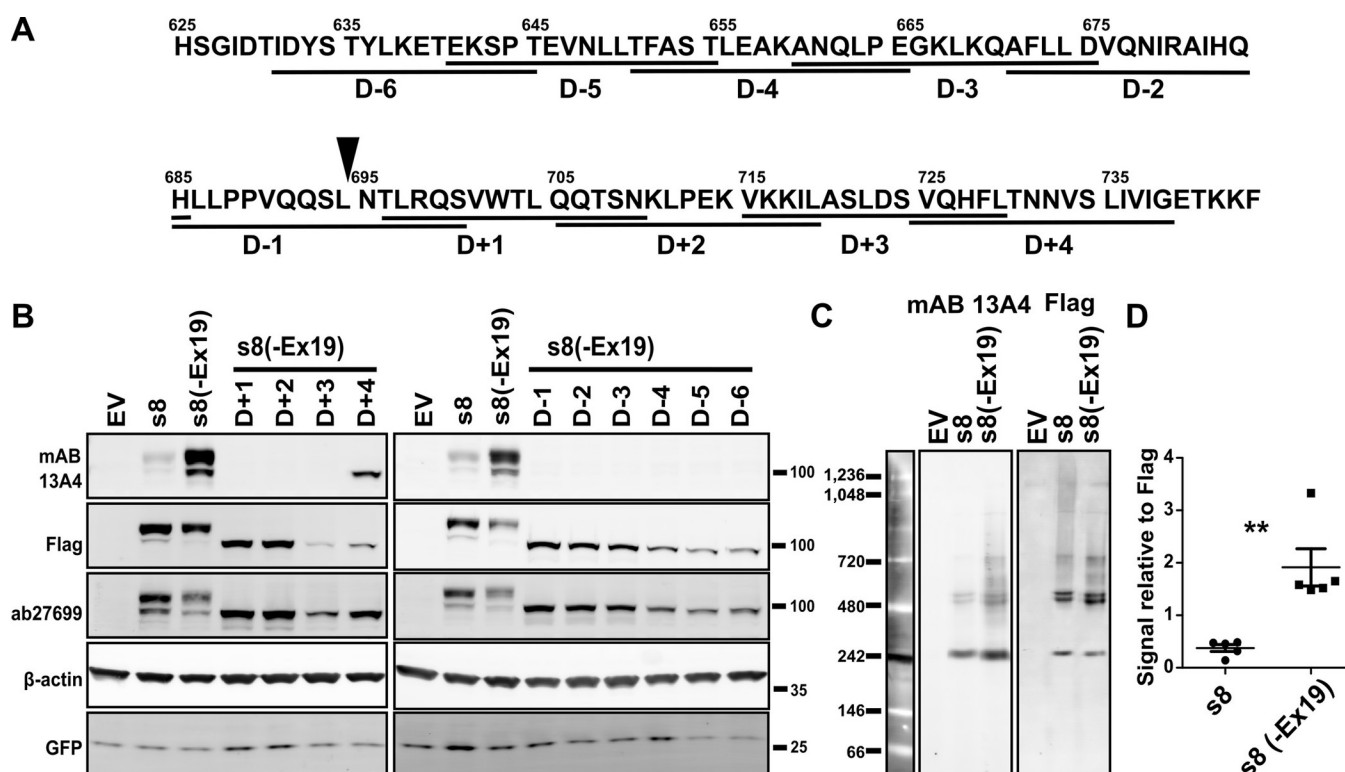

**Fig 4. Mapping of the mAB 13A4 epitope. A)** Sequence of the third extracellular domain. Underlining shows the positions of the deletions used for epitope mapping. The deletion variants were generated starting from the s8 clone lacking exon 19, s8(-Ex19). Solid triangle above the sequence shows the position where exon 19 is inserted in the photoreceptor specific isoform. **B)** Western blot analysis of PROM1 deletion mutants expressed in N2a cells using mAB 13A4 and Flag-tag antibodies. All mutants with exception of D+4 resulted in loss of the mAB 13A4 epitope. Transfection with empty pcDNA3.1 vector (EV) was used as a negative control. All transfections were spiked with a vector expressing GFP to control for transfection efficiency. **C)** Native gel Western blot analysis using mAB 13A4 and Flag-tag antibodies of N2a cell lysate transfected with either an empty vector (EV), s8, or s8(-Ex19). **D)** Quantification of mAB 13A4 signals relative to the signal from the Flag-tag antibody in native gel electrophoresis western blot. Dots represent individual data points. Line and error bars the mean and the standard error of the mean (n = 5). Unpaired Student's t-test was used to assess the statistical significance and the p-values were less than 0.01 (indicated by "**").

of exon 19 in the third extracellular domain and the skipping of two exons in the cytoplasmic C-terminal domain. The structures predicted by RobeTTa for the s8 and s2 proteins were in excellent agreement with each other and with the structure of the human PROM1 predicted by Alpha Fold [28]. In the predicted PROM1 structure, the second and third extracellular domain each form two antiparallel alpha helices that are continuous with the adjacent transmembrane domains. The alpha helices formed by the second and third extracellular domains and the adjacent transmembrane domains are packed in a four helix bundle (Figs 5B and S6). Inclusion of exon 19 lengthens the second helix of the third extracellular domain causing a kink in the bundle (Figs 5B and S6). Mapping the positions of the deletions that disrupted the mAB 13A4 epitope on the PROM1 structure showed that they were located towards the middle portion and the tip of the helical bundle. The D+4 mutation, which was the only one that did not result in loss of the mAB 13A4 epitope, was the furthest from the tip of the helical bundle. Based on the positions of the deleted segments and the six amino acids encoded by exon 19 in the PROM1 structure we made three predictions: (i) deleting six amino acids adjacent to exon 19 (Del AA 6) in the photoreceptor-specific PROM1 isoform should shorten the helix compensating the inclusion of exon 19, and restore the mAB 13A4 epitope (Fig 5B, colored dark blue on the structure of s8); (ii) Deletion of a 15 amino acid segment (D-7) in the third

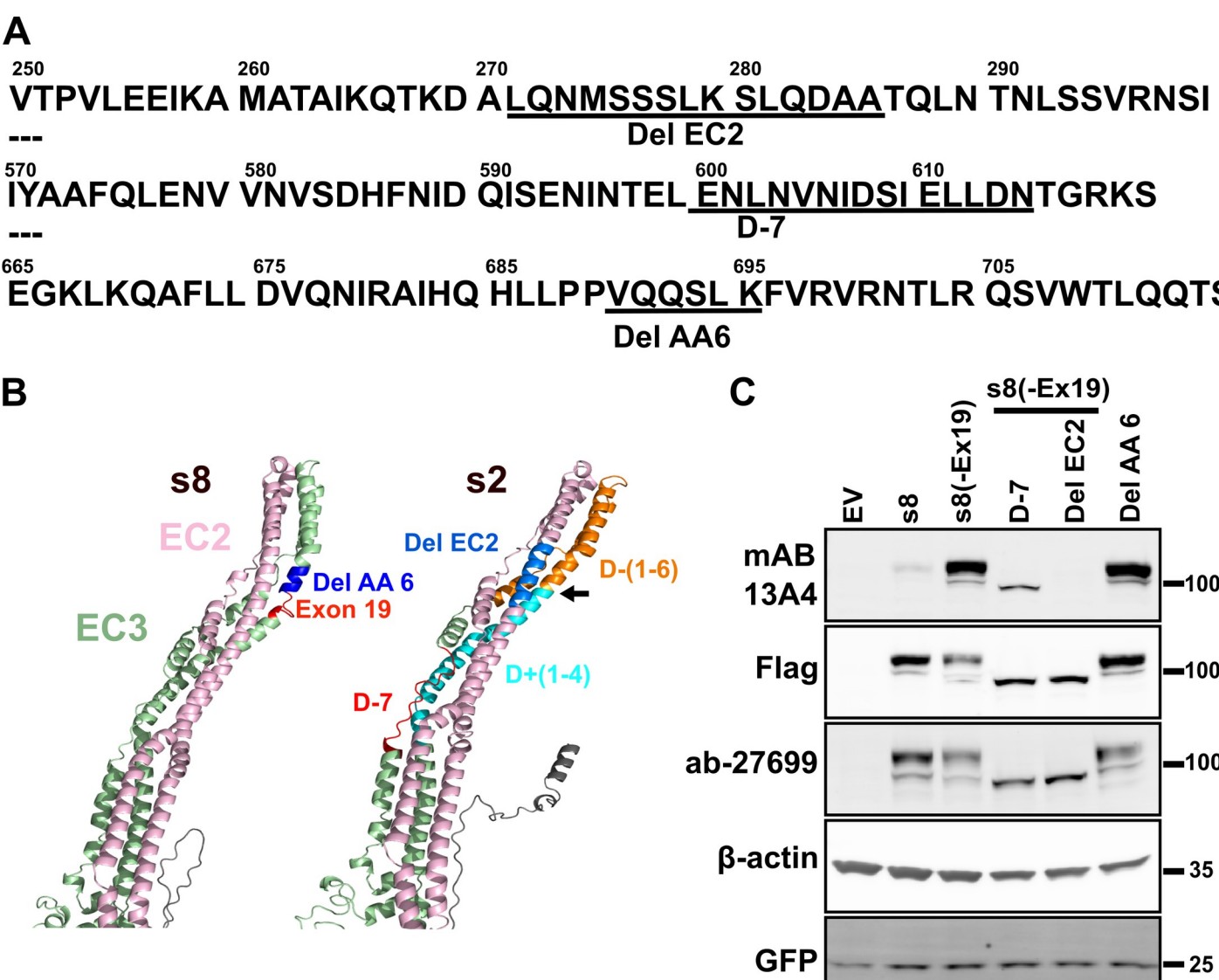

**Fig 5. Computationally derived tertiary structure of PROM1 predicts the effect of mutations on the mAB 13A4 epitope. A)** Sequence segments from the PROM1 protein with the amino acids deleted in clones Del EC2, D-7 and Del AA6 underlined. **B)** Partial structure of PROM1 isoforms s8 and s2 showing the positions of the segments deleted in the experiments on Fig 4 and on panel C of this Fig on the tertiary structure of PROM1. Exon 19 is shown in red on the structure of s8. The deletion Del AA6 analyzed on panel C is shown in blue on the structure of s8. On the structure of s2 cyan color indicates deletions D-1 through D-6 and orange indicates deletions D+1 through D+4 from the experiments shown on Fig 4. Also on the structure of s2, the positions of the deletions D-7 and EC2 analyzed on panel C are shown in red and sky blue, respectively. Arrow points to the position of the excluded exon 19 in s2. **C)** Western blot analysis of PROM1 deletion mutants expressed in N2a cells using mAB 13A4 and Flag-tag antibodies. All transfections were spiked with a vector expressing GFP to control for transfection efficiency.

extracellular domain opposite to D+4, should retain the mAB 13A4 epitope due to its distance from the location of the epitope in the upper half of the helical bundle (Fig 5B, colored red on the structure of s2); (iii) Deletion of a 15 amino acid segment (Del EC 2) in the upper half of the second extracellular domain of s8(-Ex19) should result in loss of the mAB 13A4 epitope (Fig 5B, colored sky blue on the structure of s2). The predicted structures of the Del AA 6 and D-7 mutants lacked the kink at the tip of the helix bundle that was present in the PROM1 s8 protein isoform (S5 Fig). The Del EC 2 mutant grossly distorted the helix bundle (S6 Fig). All three predictions proved to be accurate when the proteins expressed from the corresponding

clones were analyzed by Western blot (Fig 5C). Deletion of the six amino acid long segment recovered the mAB 13A4 epitope in the photoreceptor specific s8 isoform. Conversely, the deletion in the second extracellular domain of the protein encoded by the s8(-Ex19) clone resulted in complete loss of the epitope. Finally, deletion D-7 in s8(-Ex19) preserved the epitope, although the affinity of mAB 13A4 was reduced. The results of the structure directed mutagenesis provide further support for mAB 13A4 recognizing a structural epitope. In addition, our results demonstrate the utility of the modeled PROM1 structures.

## Discussion

To reliably interpret results from techniques that employ antibodies it is essential to know the antibody epitope, its specificity and its affinity. While mapping the antibody epitope may be important, it is usually not considered necessary as long as specificity to the target can be demonstrated. Such practice leaves a gap that in certain cases can have significant impact on interpreting experimental results as we show here for mAB 13A4. mAB 13A4 is widely used to detect the mouse PROM1 protein because of its excellent specificity and the lack of alternatives with comparable performance. As of the time of writing of this article, there are over 300 publications in Google Scholar citing the 13A4 antibody in the context of PROM1. Here we show that the mAB 13A4 antibody recognizes a structural epitope and its affinity for naturally occuring PROM1 isoforms can vary dramatically. When used to measure PROM1 levels in the retina, mAB 13A4 underestimated the protein amount by a factor of five compared to ab27699. In our hands deglycosylation of PROM1 reduced the signal produced by mAB 13A4 making the detection on western blot of the unglycosylated s8 isoform unreliable. It remains to be determined if a sugar moiety contributes to the epitope of the antibody or lack of glycosylation results in inefficient refolding of the protein.

Interestingly, the reactivity of the AC133 antibody, a reagent commonly used to detect the human PROM1 protein, is reduced during cancer stem cell differentiation without corresponding loss of the protein [29]. This loss of reactivity is proposed to be due to masking of the epitope by either changes in the PROM1 structure or by PROM1 protein interactors. How often do antibodies recognize structural antigens? Barlow et al. estimate that less than 10% of antigens are continuous and most antigens are formed by residues that are brought in proximity to each other by the protein structure [30]. At the time of writing of this manuscript the Immune Epitope Database (www.iedb.org) lists 6070 discontinuous and 202950 linear peptide B-cell epitopes [31]. The prediction by Barlow et al. is not necessarily in contradiction with the Immune Epitope Database records as many of the linear peptides are likely to form a secondary structure and the epitopes may in fact be discontinuous. Furthermore, the convenience of synthetic peptides or protein fragments as immunogenes is biasing against structural epitopes. Both mAB 13A4 and AC133 antibodies were raised against full length native proteins in the context of whole cell extracts or intact cells.

To determine the exact mAB1 13A4 epitope unequivocally will require determining the structure of the PROM1—mAB 13A4 complex, which is beyond the scope of the current work. Nevertheless, we show that mAB 13A4 is a useful reagent for detecting perturbation in PROM1 structure as changes to the PROM1 sequence that were hundreds of amino acids apart abolished the mAB 13A4 epitope. Furthermore we created three mutations in PROM1 guided by the computational model of its structure. The effect of these mutations on the mAB 13A4 epitope was in line with the predicted structure, providing empirical evidence for its validity. Finally, we demonstrate that under native conditions PROM1 can form higher order complexes. Further research will be needed to determine the composition of these complexes.

If the complexes prove to be PROM1 oligomers they can provide a possible path towards understanding the dominance of Prom1 mutations in patients with cone-rod dystrophy.

## Supporting information

**S1 Table. Primer sequences.**
(XLSX)

**S1 Fig. Linearity of western blot quantification. A)** Western blot replicates. Serial dilution of extract expressing clone s8(-Ex19) with extract from cells transfected with an empty vector was probed by mAB 13A4. **B)** Plot showing scaled normalized signal intensities for each replicate and linear regression with 95% confidence interval. $R^2 = 0.93$, p-value = $1.8^*10^{-12}$.
(TIF)

**S2 Fig. Gel images of replicates related to Fig 1A and 1B.** Boxes denote the parts of the images used in preparing Fig 1A.
(TIF)

**S3 Fig. Gel images of replicates related to Fig 3B and 3C.** Boxes denote the parts of the images used in preparing Fig 3B.
(TIF)

**S4 Fig. Deglycosylation of PROM1 deletion mutants.** Lysates from N2a cells were treated with deglycosylation mix II (NEB) and analyzed on western blot next to untreated controls. The blots were probed with mAB 13A4 and anti-Flag antibodies as indicated.
(TIF)

**S5 Fig. Native gel electrophoresis of PROM1 clones s8 and s8(-Ex19).** The proteins were resolved by native blue electrophoresis, transferred to PVDF membranes, and probed with mAB 13A4 and anti-Flag antibodies as indicated. Lanes containing the size standard were cut from the membranes after the transfer and stained with Ponceau S. The size standard lanes and probed membranes were imaged on Typhoon Phosphorimager (GE Healthcare).
(TIF)

**S6 Fig. Protein structure predictions for PROM1 isoform s8 and deletion clones.** Extracellular domains 2 (EC2) and 3 (EC3) are indicated on the predicted structures by green and pink color respectively. Exon 19 and the amino acids deleted in Del AA6 are colored on the structure of the s8 isoform in red and blue, respectively. The deletions for clones Del EC2 and D-7 are colored on the structure of s8(-Ex19) in light blue and red, respectively. In structures that do not contain exon 19 arrows indicate the position of the junction between exons 18 and 20. The structures of s8 and Del EC2 have pronounced kinks near the top of the bundle when compared to the structures of s8(-Ex19). Del AA6, and Del EC2.
(TIF)

**S1 Raw images.**
(PDF)

## Acknowledgments

We are grateful to Dr. Maxim Sokolov and Dr. Douglas Kolson for their assistance with native gel electrophoresis.

## Author Contributions

**Conceptualization:** Fatimah Matalkah, Visvanathan Ramamurthy, Peter Stoilov.

**Data curation:** Fatimah Matalkah, Peter Stoilov.

**Formal analysis:** Fatimah Matalkah, Peter Stoilov.

**Funding acquisition:** Peter Stoilov.

**Investigation:** Fatimah Matalkah.

**Methodology:** Fatimah Matalkah, Peter Stoilov.

**Project administration:** Peter Stoilov.

**Resources:** Scott Rhodes, Visvanathan Ramamurthy.

**Supervision:** Visvanathan Ramamurthy, Peter Stoilov.

**Visualization:** Fatimah Matalkah, Peter Stoilov.

**Writing – original draft:** Fatimah Matalkah.

**Writing – review & editing:** Visvanathan Ramamurthy, Peter Stoilov.

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
