## [Decision Letter · Decision Letter 0]

28 Jul 2022

PONE-D-22-16255The mAB 13A4 monoclonal antibody to the mouse PROM1 protein recognizes a structural epitopePLOS ONE

Dear Dr. Stoilov,

Thank you for submitting your manuscript to PLOS ONE. After careful consideration, we feel that it has merit but does not fully meet PLOS ONE’s publication criteria as it currently stands. Therefore, we invite you to submit a revised version of the manuscript that addresses the points raised during the review process.

We look forward to receiving your revised manuscript.

Kind regards,

Hemant Khanna

Academic Editor

PLOS ONE

Journal Requirements:

"National Institutes of Health, National Eye Institute grant number 2R01EY025536 to P.S. and V.M. (https://www.nei.nih.gov/)

West Virginia University Health Sciences Center Office of Research and Graduate Education bridge funding (no grant number)  to P.S. (https://health.wvu.edu/research-and-graduate-education/)"

Reviewers' comments:

Reviewer's Responses to Questions

**Comments to the Author**

1. Is the manuscript technically sound, and do the data support the conclusions?

Reviewer #1: Yes

Reviewer #2: Partly

2. Has the statistical analysis been performed appropriately and rigorously? 

Reviewer #1: Yes

Reviewer #2: Yes

3. Have the authors made all data underlying the findings in their manuscript fully available?

Reviewer #1: Yes

Reviewer #2: Yes

4. Is the manuscript presented in an intelligible fashion and written in standard English?

Reviewer #1: Yes

Reviewer #2: Yes

5. Review Comments to the Author

Reviewer #1: Matalkah et al. investigated the character of the 13A antibody, which recognizes CD133/Prom1. They found that the 13A4 recognises the structure of the Prom1 protein, rather than a single continuous sequence. Particularly the Prom1 conveying the exon 19 reduces the affinity to the 13A4 antibody. They conclude that the isofom composition needs to be considered in using this antibody because some of the isoforms may not be detected by this antibody.

The analyses have been overall done well, and provide a good piece of information for usage of the CD133/Prom1 antibody. I would support the publication of this manuscript. I have a few comments.

(1) p20. Line 4 tertiary for PROM1; Do the authors mean “tertiary structure”?

(2) Is it possible to predict the ratio of the different isoforms of Prom1 by using two different antibodies of ab27699 and m13A4? Can the authors discuss based on the data of Fig 2B?

Reviewer #2: Prom1 is a factor with an important role in the maintenance of syria and is involved in retinal degeneration; several splicing variants of Prom1 exist and are thought to have cell type-dependent regulatory mechanisms, such as being strongly expressed in neurons and other cells. In this paper, we compare mAB13A4 and ab27699 and find that the recognition efficiency differs during the post-developmental phase, indicating differential induction of splicing variants after development. They also claim that mAB13A4 recognizes two extracellular domains based on Western Blot (+SDS) validation.

While these results may have important implications for the analysis of Prom1 expression, there are several details that are of concern.

It is interesting to note that this occurs in signals that are recognized by antibodies in the post-developmental phase (Fig2). However, this phenomenon is a common one that can be observed in almost all genes with splicing variants. Furthermore, many of the antibodies sold by companies do not reveal the details of the recognition sites, and for these reasons, those conducting research with antibodies are always aware of the effects of splicing variants. Therefore, it is necessary to clarify the biological significance of this phenomenon, which can be approached by examining the following two details.

1. to clarify whether the difference in signal between mAB 13A4 and ab27699 detected in this study is due to the difference in splicing variants, it is necessary to examine mRNAs from P0 to P110 by PCR or other means. This would also allow us to further discuss the recognition of Exon19 (KFVRVR) by mAB 13A4, which is discussed in Fig. 3 and beyond.

2. it will be important to determine if the proteins detected in mAB 13A4 and ab27699 have any biological significance to maintain the significance of this paper. If the sequence of the splicing variant recognized by mAB 13A4 and ab27699 is revealed, e.g., by method 1, it should be shown experimentally.

In the description of Fig. 4, it says "Interestingly, under native conditions PROM1 formed higher order complexes corresponding in size to dimers and tetramers. Is this someone else's report? Or is this the content of Fig4C? You should clarify which one it is.

However, if you are referring to Fig. 4C, this data alone does not prove that it is a dimer. To show that it is a dimer or a tetramer, we need Immunoprecipitation, analysis using mutants of the amino acid sequence necessary for Prom1-Prom1 interaction, or analysis of the crystal structure of Prom1. For immunoprecipitation, for example, co-transfection of Flag-Prom1 and HE-Prom1, Immunoprecipitation with Flag and WB with HE. The data in Fig. 4C alone does not prove that Prom1 forms homodimers with each other, and the presence of heterodimers such as Prom1-X is also suspected.

The results of Fig5C are of great concern.

Western Blot (+SDS) would be expected to alter the structure of many of the proteins, especially membrane proteins, and not maintain the structure of a second or third extracellular domain. Therefore, Native-PAGE may alleviate this concern.

This paper uses 13A4 and ab27699, what is the difference between them?

13A4 is the hybridoma clone number and ab27699 is the product model number. 13A4 was purchased from ThermoFisher, so it seems correct to use the product model number for 13A4 as well. Even if ab27699 did not have a clone number, it would be misleading not to refer to 13A4 by the product model number. Or is 13A4 using a special purification method?

Although this paper focuses on 13A4, are the results obtained in ab27699 consistent with previous reports or information disclosed by the company? This should be included in Discussion.

6. PLOS authors have the option to publish the peer review history of their article (what does this mean?). If published, this will include your full peer review and any attached files.

Reviewer #1: No

Reviewer #2: No

---

## [Author Response · Author response to Decision Letter 0]

23 Aug 2022

Reviewer #1: Matalkah et al. investigated the character of the 13A antibody, which recognizes CD133/Prom1. They found that the 13A4 recognises the structure of the Prom1 protein, rather than a single continuous sequence. Particularly the Prom1 conveying the exon 19 reduces the affinity to the 13A4 antibody. They conclude that the isofom composition needs to be considered in using this antibody because some of the isoforms may not be detected by this antibody.

The analyses have been overall done well and provide a good piece of Information for usage of the CD133/Prom1 antibody. I would support the publication of this manuscript. I have a few comments.

# We thank the reviewer for the supportive evaluation of our work. Below are our responses to the specific comments. 

(1) p20. Line 4 tertiary for PROM1; Do the authors mean "tertiary structure"?

# Thank you for catching this. The sentence is now fixed and reads "... a tertiary structure of PROM1" (page 14, line 297). 

(2) Is it possible to predict the ratio of the different isoforms of Prom1 by using two different antibodies of ab27699 and m13A4? Can the authors discuss based on the data of Fig 2B?

# As shown in Figure 2 the signals from the ab27699 and mAb 13A4 antibodies can be used to compare two or more samples and determine if there is a difference in the isoform expression. While after careful calibration it may be possible to use the antibodies for measuring the isoform ratio we think such determination will be semi-quantitative at best. The reasons for that are:

The antibodies are not isoform specific. mAB 13A4 still recognizes both isoforms albeit it has reduced affinity to one of the isoforms. The second antibody, ab27699, recognizes both isoforms with similar affinity.

We have not exhaustively determined how post-synthetic modifications like glycosylation affect the affinity of the two antibodies. . 

For quantitative assessment of isoform expression on protein level we recommend using isobaric labeling mass-spectroscopy. For the benefit of the reviewer we attach below a plot representing peptide level signals for Prom1 from a mass-spectrometry data set we recently generated (10.6019/PXD030748) in the PDF version of our response.

Reviewer #2: Prom1 is a factor with an important role in the maintenance of syria and is involved in retinal degeneration; several splicing variants of Prom1 exist and are thought to have cell type-dependent regulatory mechanisms, such as being strongly expressed in neurons and other cells. In this paper, we compare mAB13A4 and ab27699 and find that the recognition efficiency differs during the post-developmental phase, indicating differential induction of splicing variants after development. They also claim that mAB13A4 recognizes two extracellular domains based on Western Blot (+SDS) validation.

While these results may have important implications for the analysis of Prom1 expression, there are several details that are of concern.

It is interesting to note that this occurs in signals that are recognized by antibodies in the post-developmental phase (Fig2). However, this phenomenon is a common one that can be observed in almost all genes with splicing variants. Furthermore, many of the antibodies sold by companies do not reveal the details of the recognition sites, and for these reasons, those conducting research with antibodies are always aware of the effects of splicing variants. Therefore, it is necessary to clarify the biological significance of this phenomenon, which can be approached by examining the following two details.

# We thank the reviewer for carefully reviewing our work. We agree that ascribing a biological function to the alternative isoforms of Prom1 would broaden the appeal of our work. Unfortunately we do not think inclusion of exon 19 is likely to have an effect on the function of Prom1, because of its loss in primates (Figure 1). The significance of our work is that it describes an undocumented behaviour of a widely used reagent. While we agree that the potential of alternative splicing to change the antigenic properties of proteins should be routinely considered by researchers, we find little evidence in the literature that this is the case. In the case of mAB 13A4 this phenomenon remained unnoticed for 25 years after the initial publication, despite the relatively broad use of the antibody. Below are our responses to the specific comments.

1. to clarify whether the difference in signal between mAB 13A4 and ab27699 detected in this study is due to the difference in splicing variants, it is necessary to examine mRNAs from P0 to P110 by PCR or other means. This would also allow us to further discuss the recognition of Exon19 (KFVRVR) by mAB 13A4, which is discussed in Fig. 3 and beyond.

# The switch in splicing of Prom1 in the postnatal retina was shown previously by us. We would like to direct the reviewer to Murphy et al., PLoS genetics 2016, Figure 3 B where we have analyzed the splicing of Prom1 exon 19 in the retina by RT-PCR between postnatal day 0 and day 16. A copy of the figure is included in the PDF version of our response. Inclusion levels of exon 19 switch from mostly skipped at postnatal day 0 to near complete inclusion by postnatal day 12-16. Inclusion levels remain constant after postnatal day 16 based on RNASeq data from the same article and multiple RNASeq datasets published by others. The work by Murphy et al is referenced in this context on page 11 lines 216-219 in the results section. 

2. it will be important to determine if the proteins detected in mAB 13A4 and ab27699 have any biological significance to maintain the significance of this paper. If the sequence of the splicing variant recognized by mAB 13A4 and ab27699 is revealed, e.g., by method 1, it should be shown experimentally.

# Prom1 has a well established role in stem cells, cancer development and vision. As we review in the introduction mutations in Prom1 are commonly associated with retinal degenerations. We do not think inclusion or skipping of exon 19 has an effect on the function of the Prom1 protein, because the exon is lost in primates (Figure 1B). While the biological significance of Prom1 isoforms containing exon 19 is questionable, our work is significant technically because of the wide use of the mAB 13A4 antibody in cancer and vision research. mAB 13A4 is the most often used reagent for detecting the mouse Prom1 protein with over 300 publications using it. We believe that it is important for investigators to know that the affinity of the antibody is affected by alternative splicing, especially when it is being used for quantitative assessment of Prom1 levels across tissues or developmental time points. 

3. In the description of Fig. 4, it says "Interestingly, under native conditions PROM1 formed higher order complexes corresponding in size to dimers and tetramers. Is this someone else's report? Or is this the content of Fig4C? You should clarify which one it is.

However, if you are referring to Fig. 4C, this data alone does not prove that it is a dimer. To show that it is a dimer or a tetramer, we need Immunoprecipitation, analysis using mutants of the amino acid sequence necessary for Prom1-Prom1 interaction, or analysis of the crystal structure of Prom1. For immunoprecipitation, for example, co-transfection of Flag-Prom1 and HE-Prom1, Immunoprecipitation with Flag and W.B. with HE. The data in Fig. 4C alone does not prove that Prom1 forms homodimers with each other, and the presence of heterodimers such as Prom1-X is also suspected.

# We agree with the reviewer that the data does not prove that these bands are dimers or tetramers. We removed the reference to dimers and tetramers in this sentence (page 14 line 292) and in the Discussion section (page 18, lines 393-394). The two sentences now only point to the existence of higher order complexes without speculating about their composition.

4. The results of Fig5C are of great concern. Western Blot (+SDS) would be expected to alter the structure of many of the proteins, especially membrane proteins, and not maintain the structure of a second or third extracellular domain. Therefore, Native-PAGE may alleviate this concern.

# The reviewer is correct that SDS will denature the protein. We disagree with the reviewer that this is a problem for Figure 5. On the blot shown on Figure 5 we have two samples that show gain of signal relative to the parental s8 isoform: s8(-Ex19) and Del AA6 mutant. Thus, loss of signal due to denaturation of the proteins with SDS is not an issue. The reason is that Prom1 renatures as SDS is removed during the transfer and/or blocking of the membrane. This is not an uncommon phenomenon.

5. This paper uses 13A4 and ab27699, what is the difference between them? 13A4 is the hybridoma clone number and ab27699 is the product model number. 13A4 was purchased from ThermoFisher, so it seems correct to use the product model number for 13A4 as well. Even if ab27699 did not have a clone number, it would be misleading not to refer to 13A4 by the product model number. Or is 13A4 using a special purification method?

# The 13A4 is how this antibody was first described and how it is commonly referred to in the literature. The mAB 13A4 name is also how the antibody is recorded in the CiteAB database. In addition, the 13A4 antibody is also available from Sigma-Aldrich under a different part number. Consequently, it will be confusing if we use the vendor part number. In contrast, ab27699 is only available from AbCam and has no other designations.

6. Although this paper focuses on 13A4, are the results obtained in ab27699 consistent with previous reports or Information disclosed by the company? This should be included in Discussion.

# The antibody is consistent with the vendor’s specifications. We confirmed the specificity of the antibody using a mouse model that does not express Prom1 protein (Figure 2C). As can be seen from the figures in our manuscript the ab27699 antibody produces a higher background compared to mAB 13A4, at least in our hands. In the literature we could only find use of the antibody for IHC/IF. We cannot comment on the performance of the antibody in these applications.

---

## [Decision Letter · Decision Letter 1]

8 Sep 2022

The mAB 13A4 monoclonal antibody to the mouse PROM1 protein recognizes a structural epitope

PONE-D-22-16255R1

Dear Dr. Stoilov,

We’re pleased to inform you that your manuscript has been judged scientifically suitable for publication and will be formally accepted for publication once it meets all outstanding technical requirements.

Kind regards,

Hemant Khanna

Academic Editor

PLOS ONE

Additional Editor Comments (optional):

Reviewers' comments:

Reviewer's Responses to Questions

**Comments to the Author**

1. If the authors have adequately addressed your comments raised in a previous round of review and you feel that this manuscript is now acceptable for publication, you may indicate that here to bypass the “Comments to the Author” section, enter your conflict of interest statement in the “Confidential to Editor” section, and submit your "Accept" recommendation.

Reviewer #2: All comments have been addressed

2. Is the manuscript technically sound, and do the data support the conclusions?

Reviewer #2: Yes

3. Has the statistical analysis been performed appropriately and rigorously? 

Reviewer #2: Yes

4. Have the authors made all data underlying the findings in their manuscript fully available?

Reviewer #2: Yes

5. Is the manuscript presented in an intelligible fashion and written in standard English?

Reviewer #2: Yes

6. Review Comments to the Author

Reviewer #2: Reviewer #2: Prom1 is a factor with an important role in the maintenance of syria and is involved in retinal degeneration; several splicing variants of Prom1 exist and are thought to have cell type-dependent regulatory mechanisms, such as being strongly expressed in neurons and other cells. In this paper, we compare mAB13A4 and ab27699 and find that the recognition efficiency differs during the post-developmental phase, indicating differential induction of splicing variants after development. They also claim that mAB13A4 recognizes two extracellular domains based on Western Blot (+SDS) validation.

While these results may have important implications for the analysis of Prom1 expression, there are several details that are of concern.

It is interesting to note that this occurs in signals that are recognized by antibodies in the post-developmental phase (Fig2). However, this phenomenon is a common one that can be observed in almost all genes with splicing variants. Furthermore, many of the antibodies sold by companies do not reveal the details of the recognition sites, and for these reasons, those conducting research with antibodies are always aware of the effects of splicing variants. Therefore, it is necessary to clarify the biological significance of this phenomenon, which can be approached by examining the following two details.

# We thank the reviewer for carefully reviewing our work. We agree that ascribing a biological function to the alternative isoforms of Prom1 would broaden the appeal of our work. Unfortunately we do not think inclusion of exon 19 is likely to have an effect on the function of Prom1, because of its loss in primates (Figure 1). The significance of our work is that it describes an undocumented behaviour of a widely used reagent. While we agree that the potential of alternative splicing to change the antigenic properties of proteins should be routinely considered by researchers, we find little evidence in the literature that this is the case. In the case of mAB 13A4 this phenomenon remained unnoticed for 25 years after the initial publication, despite the relatively broad use of the antibody. Below are our responses to the specific comments.

1. to clarify whether the difference in signal between mAB 13A4 and ab27699 detected in this study is due to the difference in splicing variants, it is necessary to examine mRNAs from P0 to P110 by PCR or other means. This would also allow us to further discuss the recognition of Exon19 (KFVRVR) by mAB 13A4, which is discussed in Fig. 3 and beyond.

# The switch in splicing of Prom1 in the postnatal retina was shown previously by us. We would like to direct the reviewer to Murphy et al., PLoS genetics 2016, Figure 3 B where we have analyzed the splicing of Prom1 exon 19 in the retina by RT-PCR between postnatal day 0 and day 16. A copy of the figure is included in the PDF version of our response. Inclusion levels of exon 19 switch from mostly skipped at postnatal day 0 to near complete inclusion by postnatal day 12-16. Inclusion levels remain constant after postnatal day 16 based on RNASeq data from the same article and multiple RNASeq datasets published by others. The work by Murphy et al is referenced in this context on page 11 lines 216-219 in the results section.

→Thank you for directing me to the reference. 　As someone who mainly studies retinas, I assume that I do not know the detailed Prom1 behavior during the developmental stages, so I appreciate these changes.

2. it will be important to determine if the proteins detected in mAB 13A4 and ab27699 have any biological significance to maintain the significance of this paper. If the sequence of the splicing variant recognized by mAB 13A4 and ab27699 is revealed, e.g., by method 1, it should be shown experimentally.

# Prom1 has a well established role in stem cells, cancer development and vision. As we review in the introduction mutations in Prom1 are commonly associated with retinal degenerations. We do not think inclusion or skipping of exon 19 has an effect on the function of the Prom1 protein, because the exon is lost in primates (Figure 1B). While the biological significance of Prom1 isoforms containing exon 19 is questionable, our work is significant technically because of the wide use of the mAB 13A4 antibody in cancer and vision research. mAB 13A4 is the most often used reagent for detecting the mouse Prom1 protein with over 300 publications using it. We believe that it is important for investigators to know that the affinity of the antibody is affected by alternative splicing, especially when it is being used for quantitative assessment of Prom1 levels across tissues or developmental time points.

→I understand that the functional significance with or without Ex19 is not important. I support the contributor's assertion.

3. In the description of Fig. 4, it says "Interestingly, under native conditions PROM1 formed higher order complexes corresponding in size to dimers and tetramers. Is this someone else's report? Or is this the content of Fig4C? You should clarify which one it is.

However, if you are referring to Fig. 4C, this data alone does not prove that it is a dimer. To show that it is a dimer or a tetramer, we need Immunoprecipitation, analysis using mutants of the amino acid sequence necessary for Prom1-Prom1 interaction, or analysis of the crystal structure of Prom1. For immunoprecipitation, for example, co-transfection of Flag-Prom1 and HE-Prom1, Immunoprecipitation with Flag and W.B. with HE. The data in Fig. 4C alone does not prove that Prom1 forms homodimers with each other, and the presence of heterodimers such as Prom1-X is also suspected.

# We agree with the reviewer that the data does not prove that these bands are dimers or tetramers. We removed the reference to dimers and tetramers in this sentence (page 14 line 292) and in the Discussion section (page 18, lines 393-394). The two sentences now only point to the existence of higher order complexes without speculating about their composition.

→Confirmed.

5. This paper uses 13A4 and ab27699, what is the difference between them? 13A4 is the hybridoma clone number and ab27699 is the product model number. 13A4 was purchased from ThermoFisher, so it seems correct to use the product model number for 13A4 as well. Even if ab27699 did not have a clone number, it would be misleading not to refer to 13A4 by the product model number. Or is 13A4 using a special purification method?

# The 13A4 is how this antibody was first described and how it is commonly referred to in the literature. The mAB 13A4 name is also how the antibody is recorded in the CiteAB database. In addition, the 13A4 antibody is also available from Sigma-Aldrich under a different part number. Consequently, it will be confusing if we use the vendor part number. In contrast, ab27699 is only available from AbCam and has no other designations.

→I understand that no other designation other than 13A4 is common and that there is no other designation for ab27699.

Overall, I support the publication of this paper.

I apologize for partially misleading you as I have not studied Prom1 as my main focus.

I understood that the content of the paper is of greater benefit to researchers who are dealing with Prom1 than ocular research. I think such a report would be useful in the current situation, where detailed information on antibodies is often not made public by companies and researchers need to find out the details of the antibodies themselves.

I agree that the presence of Ex19 in primates has no biological significance because Prom1 is lost in primates, but I was very interested in the evolutionary significance of the need to lose Ex19 in primates in the first place. They were lost by chance and there may be no significance in losing Ex19, but I thought it would be interesting if the presence or absence of Ex119 was involved in the structure of the primate eye, which has a unique structure compared to other organisms.

7. PLOS authors have the option to publish the peer review history of their article (what does this mean?). If published, this will include your full peer review and any attached files.

Reviewer #2: No

---

## [Editor Report · Acceptance letter]

29 Sep 2022

PONE-D-22-16255R1 

The mAB 13A4 monoclonal antibody to the mouse PROM1 protein recognizes a structural epitope 

Dear Dr. Stoilov:

I'm pleased to inform you that your manuscript has been deemed suitable for publication in PLOS ONE. Congratulations! Your manuscript is now with our production department. 

Kind regards, 

on behalf of

Dr. Hemant Khanna 

Academic Editor

PLOS ONE